# Synthesis and Optical Properties of a Novel Hybrid Nanosystem Based on Covalently Modified nSiO_2_ Nanoparticles with a Curcuminoid Molecule

**DOI:** 10.3390/nano14121022

**Published:** 2024-06-13

**Authors:** Nicole Parra-Muñoz, Valentina López-Monsalves, Rodrigo Espinoza-González, Daniel Aravena, Nancy Pizarro, Monica Soler

**Affiliations:** 1Department of Chemical Engineering, Biotechnology and Materials, Faculty of Physical and Mathematical Sciences, Universidad de Chile, Santiago 8370456, Chileroespino@ing.uchile.cl (R.E.-G.); 2Centro de Materiales para la Transición y Sostenibilidad Energética, Comisión Chilena de Energía Nuclear, Ruta 68, km 20, Santiago 7600713, Chile; 3Departamento de Química Farmacológica y Toxicológica, Facultad de Ciencias Químicas y Farmacéuticas, Universidad de Chile, Santiago 8380494, Chile; 4Departamento de Química de los Materiales, Facultad de Química y Biología, Universidad de Santiago de Chile (USACH), Casilla 40, Correo 33, Santiago 9170002, Chile; daniel.aravena.p@usach.cl; 5Departamento de Ciencias Químicas, Facultad de Ciencias Exactas, Universidad Andrés Bello, Viña del Mar 2520000, Chile; npizarro@unab.cl

**Keywords:** curcuminoids, grafted SiO_2_ nanoparticles, photophysical properties, solvent effects, hybrid nanosystem

## Abstract

A new curcuminoid molecule (**3**) has been designed and synthesized, containing a central -(CH_2_)_2_-COOH chain at the α carbon of the keto-enol moiety in the structure. The carboxylic acid group is added to react with exposed amino groups on silica oxide nanoparticles (nSiO_2_), forming an amide bond to attach the curcuminoid moiety to the nSiO_2_ covalently. The Kaiser test quantifies the functionalization degree, yielding 222 μmol of curcuminoid per gram of nanoparticles. The synthesized hybrid nanosystem, nSiO_2_-NHCO-CCM, displays significant emission properties, with a maximum emission at 538 nm in dichloromethane, similar to curcuminoid **1** (without the central chain), which emits at 565 nm in the same solvent. Solvent-induced spectral effects on the absorption and emission bands of the new hybrid nanosystem are confirmed, similar to those observed for the free curcuminoid (**1**). The new nanosystem is evaluated in the presence of kerosene in water, showing an emission band at 525 nm as a detection response. The ability of nSiO_2_-NHCO-CCM to change its fluorescence when interacting with kerosene in water is notable, as it overcomes the limitation caused by the insolubility of free curcuminoid **1** in water, allowing for the exploitation of its properties when connected to the water-stable nanosystem for future detection studies.

## 1. Introduction

Research on curcuminoid molecules has its origin in curcumin. Since the 19th century, the scientific community has maintained a constant interest in curcumin due to its diverse properties. As understanding of their properties has deepened, numerous derivatives of this molecule, known as curcuminoids, have been designed and synthesized [1,2] Curcuminoids (CCMs) constitute a family of molecules that share the basic structure of curcumin, but have specific structural modifications for various applications, ranging from cancer and Alzheimer’s [3] treatments to solar energy generation [4,5] and colorimetric or fluorometric sensors for environmental contaminants [6,7,8]. The distinctive structure of curcuminoids includes a conjugated chain of seven carbon atoms, with aromatic rings at both ends that can accommodate one or more functional groups. At the core of this conjugated chain, there is a β-diketone unit in a tautomeric equilibrium, where both diketo and keto-enol species coexist. Although the predominance of one of the two species depends on factors such as solvent type, temperature, and solution pH, it has been documented that the predominant species is the keto-enol form in curcuminoids [9]. This preference is due to the ability of this configuration to form an intramolecular hydrogen bond favored by resonance, which originates from the movement of electrons through the terminal aromatic rings, the heptadiene skeleton, and the ring in the keto-enol unit [10]. Furthermore, it is feasible to carry out strategic modifications to the basic structure to fine-tune its properties. Among these modifications is the closure of the central keto-enol unit, which enhances conjugation and modifies its coordination ability [11,12]. The introduction of new groups, both at the center and at the terminal ends of the conjugated chain, is also considered, with the purpose of modulating or suppressing the optical properties of CCMs [13,14]. Similarly, through the introduction of new groups, immobilization can be achieved through chemical reactions with other materials or molecules. For example, Gogoi et al. reported a set of modified curcuminoids, named CCM-cysteine (CC) and CCM-tryptophan (CT), which they described as biomolecule-based materials for the detection of 2,4,6-trinitrophenol (picric acid (PA)) in aqueous solution [15]. On the other hand, Xu et al. designed a new curcuminoid, the sensor L (L = 1,7-bis[4′-bromobutaneyloxy-3′-methyloxy-phenyl]-1,6-heptadiene-3,5-dione), by functionalizing curcumin with bromobutane, which was described as a colorimetric and fluorescent probe capable of detecting Cu^2+^ ions over other metal ions in a phosphate-buffered saline solution [16]. However, inherent challenges such as low solubility, suppression of optical properties, or irreversibility of the probe in the detection medium may necessitate their incorporation into structures or materials to provide stability and the ability to interact effectively with the analytes of interest [17,18,19]. Several studies, predominantly focused on curcumin, have demonstrated the immobilization of curcuminoids in various materials, including halloysite nanotubes [20], cellulose (in its crystalline variants and nanofibers) [21,22], silica [23], PAMAM dendrimer [24] and graphene oxide [25], among others, to enhance the performance of curcumin properties. For example, the application of chemically modified cellulose nanofibers with curcumin in chitosan films has been reported, demonstrating an enhancement in the physical and antibacterial properties of the nanofiber curcumin, particularly in the context of its use in food packaging [21]. Ganguar et al. reported the conjugation or functionalization of curcumin with silica nanoparticles to enhance its stability in water and bioavailability in organisms [23]. However, the focus of most of these studies is curcumin, and the primary approach involves utilizing the deprotonated oxygen atom of the central β-diketone (in keto-enol equilibrium) or the -OH groups of the terminal aromatic rings of curcumin [15,20,21,22] for the functionalization. Moreover, in the detection field, curcumin exhibits several limitations, including a low emission quantum yield ranging from 0.014 to 0.075 in cyclohexane and acetonitrile, respectively, low solubility, and deficiencies in terms of photostability [26,27]. In this regard, we focus our research on the design and synthesis of an innovative set of curcuminoid molecules (**1**, **2**, and **3**), featuring aromatic ends with dimethylamine groups. These groups act as electron density donors towards the subtractive central portion of the curcuminoid molecule, the keto-enol unit, thereby enhancing certain optical properties, for example, the emission quantum yield (absolute quantum yield of curcuminoid **1** in CH_2_Cl_2_ is Φ_CH2Cl2_ = 0.69). We implemented a novel strategy for the covalent binding of the curcuminoid structure to silicon oxide nanoparticles by incorporating a central chain of -(CH_2_)_2_-COOH in the α carbon of the keto-enol unit in the curcuminoid structure (Figure 1), which was described earlier in curcumin, but not for functionalization [9]. Before the conjugation with the curcuminoid, the silicon oxide nanoparticles were functionalized with 3-aminopropyltriethoxysilane, providing them with amino terminal groups exposed on the surface capable of reacting with the carboxylic acid terminal group of the new curcuminoid molecule **3**. Silicon oxide nanoparticles (nSiO_2_), with their high surface-to-volume ratio, are ideal candidates for modification with organic molecules, allowing the acquisition of new properties while maintaining their properties in the nanoscale [28]. Our research focuses on developing new curcuminoid probes using an economical and stable platform, such as nSiO_2_, which can efficiently operate in different solvents. The primary objective of this investigation is to study how anchoring the curcuminoid to nanoparticles affects the optical behavior of the resulting nanosystem, with the goal of a future fabrication of innovative detection materials.

## 2. Materials and Methods

### 2.1. Materials 

The reactants and solvents were acquired from Merck Millipore (Darmstadt, Germany), such as (3-aminopropyl)Triethoxysilane (APTES 99%), Oxalyl chloride (99%), boric anhydride (98%), acetylacetone (acac 99%), tributyl borate (99%), n-butylamine (99.5%), lithium hydroxide (98%), (4-dimethylamine)benzaldehyde (99%), and triethylamine, as well as solvents used, ethyl acetate, dimethyl sulfoxide (DMSO), dichloromethane (CH_2_Cl_2_), acetone, tetrahydrofuran (THF), acetonitrile (CH_3_CN), and N,N-dimethylformamide (DMF), which exhibited high purity (≥99.5). SiO_2_ nanoparticles (15–20 nm) were purchased from Sigma Aldrich (Darmstadt, Germany), and Methyl-4-acetyl-5-oxohexanoate was purchased from Santa Cruz biotechnology (Dallas, TX, USA). 

### 2.2. Instruments 

The curcuminoid molecules were characterized by ^1^H-NMR spectroscopy (Multinuclear NEO Bruker Advance 400 MHz, Karlsruhe, Germany) and MALDI-TOF Microflex mass spectrometry (Bruker Daltonics Inc., Minneapolis, MN, USA) in positive ion mode with α-cyano-4-hydroxycinnamic acid and 2,5-dihydroxybenzoic acid as matrices. Prior to acquiring the spectra, the instrument was calibrated with an external standard in the *m*/*z* range of 280–1200. Common spectroscopic characterization methods for curcuminoids and nanoparticles are FTIR-ATR spectroscopy (Thermo Scientific; FTIR Nicolet iS50 spectrometer, Waltham, MA, USA) using a narrow-band mercury cadmium telluride (MCT) detector cooled by liquid nitrogen), UV–Vis absorption spectroscopy (Evolution 220 Thermo Fisher, Shanghai, China), and emission spectroscopy (Perkin Elmer LS-55 (Llantrisant, UK) and a FluoroMax-4 (Horiba Jovin Yvon, Beijing, China). The absolute emission quantum yields were determined by employing a Fluorimeter FS5 (Edinburgh Instruments, Livingston, Reino Unido (Inglaterra)) with integrating sphere, while the fluorescence lifetimes were measured using a FluoTime300 Lifetime Spectrometer (PicoQuant, Berlín, Germany). Furthermore, the nanoparticles were characterized by thermogravimetric analysis (TA Instruments TGA Q50 V20.10 Build 36, Eden Prairie, MN, USA), dynamic light scattering (DLS) and zeta potential (pZ) (Malvern ZetaSizer 3000, Malvern Instruments, Malvern, UK), high-resolution field-emission scanning electron microscope (FESEM) (Thermo Fisher, FEI, Quanta FEG 250, Hillsboro, OR, USA), and scanning transmission electron microscopy (HR-TEM; Thermo Fisher (ex FEI) Model Tecnai F20 STWIN G2 (Eindhoven, The Netherlands)).

### 2.3. Synthesis of Curcuminoids 

The **1** and **2** were synthesized with minor modifications to a previously reported method [29]. Firstly, acetylacetone (0.7 mL, 7 mmol) was used for the synthesis of **1**, whereas methyl-4-acetyl-5-oxohexanoate (1.2 mL, 7 mmol) was used for the synthesis of **2**. These compounds were stirred with boric anhydride (B_2_O_3_, 0.35 g, 5 mmol) in ethyl acetate (5.0 mL) for 30 min at 60 °C to form the correspondent boron complex (Appendix A). Next, a solution containing 4-(dimethylamine)benzaldehyde (2.5 g, 17 mmol) and tributyl borate (3.97 mL, 14 mmol) was added to each of the reaction mixtures and kept for 3 h at 60 °C. Upon completion of the reaction, the mixtures were cooled, and a solution containing n-butylamine (0.2 mL, 2 mmol) in 1 mL of ethyl acetate was added dropwise. Subsequently, the reaction mixtures were stirred overnight at room temperature. After the reaction was completed, the reaction mixtures were vacuum filtered and dried at 50 °C for 5 h. In the synthesis of **1** and **2**, a purple solid was obtained, which was subsequently suspended in a CH_3_CN/H_2_O mixture (30/70) and refluxed overnight to decompose the boron complex. This decomposition reaction was repeated a total of 2 times. Finally, both red solids were purified by recrystallization in CH_3_CN.

1,7-Bis[4-(dimethylamino)phenyl]-5-hydroxy-1,4,6-heptatrien-3-one (**1**). Yield: 72%. Melting point: 220–223 °C. FTIR-ATR (cm^−1^): 2912 (Csp^2^-H), 2814 (Csp^3^-H), 1590 (C=O ketone), 1554 (C=C), and 1521 (C-N). ^1^H-NMR (400 MHz, CDCl_3_) (ppm): 3.0 (s, 12H), 5.7 (s, 1H), 6.4 (d, 2H, J = 7.8 Hz), 6.7 (d, 4H, m), 7.5 (d, 4H, J = 4.5 Hz), 7.6 (d, 2H, J = 7.6 Mz), and 16.3 (s, 1H). MALDI-TOF (positive ionization) [M + H]^+^ calc. for [C_23_H_26_N_2_O_2_ + H]^+^: 363.46; found: 363.32.

Methyl-4-{3-[4-(dimethylamino)phenyl]prop-2-enoyl}-5hydroxy-7-{4-(dimethylamino)phenyl}-4,6-dieno heptanoate (**2**). Yield: 54%. Melting point: 234–237 °C. FTIR-ATR (cm^−1^): 2908 (Csp^2^-H), 2809 (Csp^3^-H), 1722 (C=O ester), 1588 (C=O ketone), 1546 (C=C), and 1517 (N-H). ^1^H-NMR (400 MHz, CDCl_3_): 2.3 (t, 1H, J = 8.0), 2.4 (t, 1H, J = 7.0) (ppm): 3.0 (s, 14H), 3.7 (s, 3H), 6.7 (m, 5H), 6.9 (s, 1H), 7.5 (d, 4H, J =15), 7.7 (m, 2H), and 18.0 (s, 1H). MALDI-TOF (positive ionization) [M + H]^+^ calc. for [C_27_H_32_N_2_O_4_ + H]^+^: 449.55; found: 449.56.

To a solution of curcuminoid **2** (0.2 g, 0.45 mmol) in tetrahydrofuran (40 mL), 2 mL of an aqueous solution of LiOH (0.05 g, 2.1 mmol. 1.1 M) was added. The mixture was stirred at reflux for 6 h. Once the reaction was completed, the solvent was removed under reduced pressure, and the obtained oil was redissolved in 5 mL of deionized water. Curcuminoid **3** was precipitated by adjusting the pH in the range of 5 to 5.5 with 10% HCl, subsequently filtered, and washed with deionized water (maintaining the same pH value) to obtain a yellow solid. This yellow solid, **3**, was then suspended in hot CH_2_Cl_2_ to remove impurities, filtered, and vacuum-dried at 80 °C.

4-{3-[4-(dimethylamino)phenyl]prop-2-enoyl}-5hydroxy-7-{4-(dimethylamino)phenyl}-4,6-dieno heptanoic acid (**3**). Yield: 40%. Melting point of 230–233 °C. FTIR-ATR (cm^−1^): 2901 (Csp^2^-H), 2807 (Csp^3^-H), 1662 (C=O ion carboxylate), 1592 (C=O ketone), 1546 (C=C), and 1527 (C-N). ^1^H-NMR (400 MHz, CDCl_3_) (ppm): 3.0 (m, 14H), 6.2 (d, 2H, J = 16 Hz), 6.7 (d, 4H, J = 9.0 Hz), 7.5 (m, 6H), and 12.0 (m, 1H). MALDI-TOF (positive ionization) M + H calc. for C_26_H_30_N_2_O_4_: 434.53; found: 434.27.

### 2.4. Silanization Reaction of nSiO_2_ Nanoparticles with 3-Aminopropyltriethoxysilane

Following the procedure of Karnati et al. [30] with some modifications, 1000 mg of nSiO_2_ nanoparticles (20 nm) was suspended in a 100 mL mixture of ethanol and water in a 95/5 ratio (Appendix A). The suspension was immersed in an ultrasonic bath for 10 min. After this time, 3100 μL of 3-aminopropyltriethoxysilane (APTES) was added, and the suspension was again immersed in the bath for an additional 5 min. The sample was then heated to 70 °C under stirring for 2 h. After this time, it was left stirring overnight at room temperature for 24 h. The next day, the suspension was concentrated by centrifugation and washed four times with 10 mL ethanol. The resulting solid was dried under vacuum for 24 h at 80 °C. A total amount of 0.728 g of nSiO_2_-NH_2_ was obtained.

### 2.5. Quantification of Amino Group

The quantification of amino groups functionalized onto nanoparticles was carried out using the colorimetric assay known as the Kaiser test. In this assay, ninhydrin reacts with primary amines, producing an intense blue color known as Ruhemann’s purple. The detection process begins when ninhydrin comes into contact with the primary amine. At this point, deamination occurs, resulting in the release of ammonia and carbon dioxide. Subsequently, the ammonia molecule reacts with ninhydrin, generating diketohydrin (Ruhemann’s complex) [31].

Kaiser test on nSiO_2_-NH_2_. Following the procedure of Cueto-Díaz et al. [31] with some modifications, between 5 and 10 mg of nSiO_2_-NH_2_ was taken and deposited into an amber vial. Additionally, 5 mL of ethanol and 1 mL of a 0.016 M ninhydrin solution in ethanol were added. The vial was closed and sealed with Teflon and parafilm, and the mixture was allowed to react for 30 min at a temperature of 100 °C. Subsequently, the product was cooled and centrifuged. The supernatant was employed to acquire a UV-Vis spectrum, from which the absorbance value at 579 nm was obtained, the peak absorption value of the Ruhemann’s complex. This process was carried out in triplicate. Simultaneously, a stock solution of butylamine was prepared under the same reaction conditions to construct the calibration curve. For the calibration curve generation, stock solutions of butylamine and ninhydrin in ethanol were prepared with concentrations of 0.016 M and 0.010 M, respectively. The ninhydrin solution was kept in darkness throughout the entire reaction process. Six fresh solutions were prepared from the butylamine stock solution, with concentrations ranging from 5.89 × 10^−4^ M to 1.00 × 10^−5^ M. To each of these solutions, 1.0 mL of the ninhydrin solution was added, and the volume was adjusted to 6 mL with absolute ethanol. Each solution was hermetically sealed and allowed to react at 100 °C for 30 min. Each of these solutions was subjected to absorbance spectrum acquisition in triplicate, and the absorbance value at 579 nm was utilized to construct the calibration curve.

### 2.6. Formation of Amide Bond to Prepare nSiO_2_-NHCO-CCM Systems

The synthetic strategy involves the reaction between the terminal -NH_2_ group of the silane functionalized nSiO_2_ nanoparticles and the carboxylic acid moieties of the curcuminoid molecule to form an amide bond. To achieve this new bond, the terminal acid group of the central chain is activated in the form of acyl chloride, which facilitates the nucleophilic reaction between the amino group and the carbonyl group of the curcuminoid molecule. During the study, three different quantities of **3** were tested: 10 mg, 30 mg, and 50 mg. Activation of the carboxylic acid group of the curcuminoid was carried out using oxalyl chloride [32]. First, the curcuminoid–acid was suspended in 10 mL of dry CH_2_Cl_2_. Then, 5, 15, and 25 µL of oxalyl chloride were added to the 10, 30, and 50 mg of curcuminoid molecule, respectively, in 3 drops of dry DMF. The initial, yellow-colored mixture of the curcuminoid–acid molecules turned red after the addition of oxalyl chloride. The reaction mixtures were stirred for 1 h at room temperature. After this time, the reaction mixtures were evaporated under reduced pressure. The resulting products were suspended in 3 mL of dry CH_2_Cl_2_, and the solvent was evaporated again to remove any excess unreacted oxalyl chloride. This process was repeated a total of 3 times. The remaining oily substance was then suspended in 3 mL of dry CH_2_Cl_2_ and added to 100 mg of nanoparticles previously suspended in 3 mL of dry CH_2_Cl_2_. In addition, 4, 11, and 18 µL of triethylamine were added to the suspension, respectively. These mixtures were left to react for 4 days at room temperature. After this time, the nanoparticles were concentrated and washed by centrifugation with CH_2_Cl_2_ and THF until the supernatant became clear. Different portions were used to quantify and determine the number of unreacted amino groups through the Kaiser test. The difference between the initial and final numbers of amino groups allowed us to calculate the moles of curcuminoid anchored onto the nanoparticles.

### 2.7. Ab Initio Calculations

All calculations were performed by means of the ORCA 5.0.4 program package [33]. DFT and TD-DFT results were based on the TPSSh density functional [34,35] and the Def2-TZVP basis set [36]. Solvent effects were described by the CPCM approach [37]. TD-DFT-based absorption spectra were obtained on top of TPSSh optimized geometries for each solvent, which were corroborated as local minima by the calculation of their vibrational frequencies. The effect of the solvent in the structural relaxation of **1**–**3** was tested by recalculating the absorption spectrum in different solvents, all based on the gas-phase-optimized geometry. Results were similar to the values optimized for the solvent, so the shift in the absorption maximum is related to electronic effects rather than specific geometric distortions connected to different solvents.

### 2.8. Multilinear Regression Analysis

A multilinear regression analysis for Linear Solvation Energy Relationships (LSER) was conducted using the Kamlet–Taft (1) [38,39] and Catalán (2) solvent scales [40,41].

Kamlet–Taft equation:υ_emi_ = υ_o_ + sπ* + aα + bβ(1)

Catalán equation:υ_emi_ = υ_o_ + aSA + bSB + cSdP + dSP(2)

For Equation (1), the independent variables are π*, α, and β, while, for the Catalán scale, they are SA, SB, SdP, and SP. In both analyses, the dependent variable is the wavelength maximum (in cm^−1^). The analysis was performed using the regression application in an Excel spreadsheet with a confidence level of 95%. This analysis relies on purely statistical criteria. Parameters such as sample size (N), correlation coefficient (R), standard deviation (SD), and Fisher’s index of equation reliability (F) indicate the overall quality of the correlation equation. The reliability of individual terms is evaluated using the t-statistic (t-stat) and the variance inflation factor (VIF). A robust correlation equation is characterized by high values of N, F, and t-stat, low SD values, and R and VIF values close to unity [42].

## 3. Results and Discussion

### 3.1. Chemistry of Curcuminoid Molecules

Curcuminoid molecules **1**, **2**, and **3** were synthesized via a three-step process involving boron complex formation, an Aldol condensation reaction with (4-dimethylamine)benzaldehyde, and subsequent complex breakdown to yield the respective curcuminoids. Structural elucidation relied on FTIR-ATR, ^1^H-NMR, and MALDI–TOF spectroscopy. FTIR-ATR spectra comparison (Appendix A) revealed a keto-enol equilibrium, evidenced by shifted stretching signals of the C=O bond. Curcuminoid 2 displayed a distinct stretching signal of the C=O bond in the ester group, while curcuminoid 3 showed a characteristic vibration signal of the C=O group, suggesting a carboxylate ion form. This was attributed to the synthesis process, where precipitation of the hydrolysis product occurred at pH 5–5.5, rendering curcuminoid 3 soluble due to nitrogen atom protonation. Signals for C=C and C-N were observed in all cases. In the ^1^H-NMR spectra comparison (Appendix A), curcuminoids **1** and **2** exhibited signals corresponding to the enolic proton, with distinct chemical shifts attributed to the different groups in the α carbon of the keto-enol moiety. In contrast, the enolic signal of curcuminoid **3** was not observed, likely due to proton exchange with the -COO^−^ group, causing this signal to fall outside the detection time of the NMR equipment. On the other hand, curcuminoid 1 displayed a characteristic proton signal on the α carbon between the keto-enol group. Additionally, a new signal characteristic of a carboxylic acid proton was observed in curcuminoid **3**, suggesting its presence in solution as a carboxylate ion (see Appendix A). MALDI–TOF analysis confirmed the expected ionic molecular fragments for all curcuminoids.

### 3.2. Chemistry of Nanoparticles

Silica nanoparticles (nSiO_2_) were functionalized by the reaction with 3-aminopropyltriethoxysilane (APTES), resulting in nSiO_2_-NH_2_ nanoparticles surrounded by terminal -NH_2_ groups (Figure 1b). To assess the degree of functionalization on the nanoparticles, the Kaiser colorimetric assay was employed, and, according to the assay using the calibration curve shown in Appendix A, a quantification of 238 micromoles of amino groups per gram of nanoparticles was obtained for the nSiO_2_-NH_2_ (Table 1). Then, the -NH_2_-functionalized nanoparticles (nSiO_2_-NH_2_) were used to react with the -(CH2)_2_-COOH chain of curcuminoid **3**, resulting in the formation of an amide bond. Through a nucleophilic attack, the -NH_2_ terminal group reacted with the carboxylic acid previously activated as an acyl chloride, generating the nSiO_2_-NHCO-CCM system. Three stoichiometric ratios were employed between nSiO_2_-NH_2_ and curcuminoid **3** to form the nSiO_2_-NHCO-CCM, with mass ratios of 100/10, 100/30, and 100/50, resulting in nSiO_2_-NHCO-CCM-10, nSiO_2_-NHCO-CCM-30, and nSiO_2_-NHCO-CCM-50, respectively. The nanoparticles modified with the curcuminoid molecule exhibited a pale pink color in all cases, contrasting with the white color of the powder of the original nanoparticles. Afterwards, the Kaiser colorimetric assay was performed again to quantify the remaining amino groups on the surface of the three different nSiO_2_-NHCO-CCM systems after the reaction with curcuminoid **3**. The differences between the quantifications before and after anchoring the curcuminoid **3** to the nSiO_2_-NH_2_ using the three different stoichiometries resulted in values of 118, 222, and 210 micromoles of curcuminoid molecules per gram of nanoparticles, respectively (Table 1). It can be inferred from these results that there is a significant increase in the level of functionalization when moving from the stoichiometry 100/10 to 100/30, but, with further increases (100/50), no significant changes were observed.

In Appendix A is shown the FTIR-ATR spectrum of the chemically modified nanoparticles, SiO_2_-NHCO-CCM, considering the different stoichiometries, and it was observed that their spectra are quite similar, emphasizing C=O stretching (amide), along with signals provided by the base structure of nSiO_2_-NH_2_ nanoparticles. Figure 2 presents the FTIR spectra of **3**, pristine silica nanoparticles (nSiO_2_), silica nanoparticles with 3-aminopropyltriethoxysilane functionalization (nSiO_2_-NH_2_), and nanoparticles anchored with the curcuminoid molecule (nSiO_2_-NHCO-CCM) using the 100/30 molar ratio. In the case of nSiO_2_, the stretching signal of SiO-H around 3500 cm^−1^, the symmetrical stretching vibration of the Si-O-Si bond at 1150 cm^−1^, and the bending of O-Si-O at 946 cm^−1^ were observed, along with the Si-C signal at 790 cm^−1^. Following the chemical modification of silica nanoparticles with APTES, the N-H bending signal at 1544 cm^−1^ was evident, in addition to the characteristic signals of the nanoparticles at low wavenumber. Upon analyzing the nSiO_2_-NHCO-CCM nanoparticles, a new signal was highlighted, associated with the stretching of the newly formed amide bond between the amino terminal and the curcuminoid, identified at 1610 cm^−1^.

Appendix A presents the hydrodynamic diameter distribution of the different nanostructures of nSiO_2_-NHCO-CCM using the Dynamic Light Scattering (DLS) technique compared with the pristine silica nanoparticles (nSiO_2_) in ethanol. It can be observed that all the nanosystems synthesized, nSiO_2_-NHCO-CCM, exhibited a monodisperse distribution with an average size between 388 and 518 nm. Our results indicate that the increase in stoichiometry in the synthesis of nSiO_2_-NHCO-CCM caused slight variations in their hydrodynamic radius compared to base nSiO_2_. The most noticeable variation was observed when comparing nSiO_2_-NHCO-CCM-10 and nSiO_2_-NHCO-CCM-30, which could be attributed to a higher degree of functionalization in the case of nSiO_2_-NHCO-CCM-30, increasing the probability of interactions with the solvent and expanding the hydrodynamic diameter [43]. On the other hand, when comparing SiO_2_-NHCO-CCM-30 and nSiO_2_-NHCO-CCM-50, minor differences were observed, suggesting that there may not be a significant disparity in the degree of curcuminoid functionalization. Additionally, Appendix A reports the average hydrodynamic diameter (Z) and polydispersity index (PDI) measurements of the different nanosystems. The relative standard deviation (%RSD) of the diameter of the nanostructures showed values below 8%. Additionally, the polydispersity indices (PDI) for all measurements remained below 0.511, indicating a narrow particle size distribution and highlighting the good stability of the suspension of the nanostructures. The Z potential of the nanostructures was 37.8 mV for nSiO_2_-NHCO-CCM-10, and 40.9 and 41.3 mV for nSiO_2_-NHCO-CCM-30 and nSiO_2_-NHCO-CCM-50, respectively, while the potential for pristine nanoparticles was −27.5 mV. Pristine nanoparticles exhibit a negative surface charge attributed to the presence of hydroxyl groups in their structure. As curcuminoid molecules are functionalized, the surface charge increases due to the structure of the curcuminoids. These compounds contain nitrogen atoms in their structure, which can be protonated to enhance the surface charge [44,45]. These results align with the observed hydrodynamic diameters. The most notable change of Z potential was observed between nSiO_2_-NHCO-CCM-10 and nSiO_2_-NHCO-CCM-30, while a less pronounced change was observed when comparing SiO_2_-NHCO-CCM-30 and nSiO_2_-NHCO-CCM-50 (Appendix A). Therefore, in nSiO_2_-NHCO-CCM-10, there may be a lower amount of curcuminoid functionalized compared with the other two systems.

The thermogravimetric analysis (TGA) of all the nSiO_2_-NHCO-CCM nanostructures was compared with the pristine nSiO_2_ nanoparticles and curcuminoid **3**, as shown in Appendix A. In each hybrid nanosystem, an initial mass loss was observed with a TGA curve peak between 100 and 110 °C, attributed to the desorption of adsorbed water on the surface of the system, corresponding to approximately 4% of the mass. This phenomenon was observed in all nanoparticles but was not observed in the isolated curcuminoid molecule, which exhibited a mass loss around 200 °C, indicative of a behavior consistent with an organic compound. The mass losses attributed to the decomposition of organic molecules in the nanostructures were observed in the range of 250–350 °C, indicating the presence of functionalized organic compounds distinct from the decomposition pattern of the free curcuminoid molecule. The weight loss after 350 °C decreased, which can be assigned to the formation of pure nSiO_2_ particles.

The morphology of the nanoparticles, both pristine and of the nSiO_2_-NHCO-CCM nanostructures, was examined using FE-SEM and TEM. As illustrated in Figure 3, all obtained nanoparticles exhibited irregular morphologies with a tendency toward sphericity. In the micrographs, individual nanoparticles of similar sizes with smooth surfaces were identified. Given that the goal is to investigate the effect of functionalization with the curcuminoid molecule on the nanoparticle properties and the progress towards developing a detection system, the lack of uniformity in the nanoparticles is not considered a negative effect for this study. On the contrary, it contributes to a higher surface area, enhancing the potential of nanostructures as optical detectors with an increased surface area for detection. Through this analysis, it was observed that the morphology of nSiO_2_ nanoparticles undergoes no significant changes when functionalized with the curcuminoid molecule (nSiO_2_-NHCO-CCM). The amorphous form of the pristine nanoparticles was maintained, along with an approximate size of 20 nm in all cases (Appendix A). Additionally, Figure 3 presents two inserts corresponding to TEM images of nSiO_2_ and nSiO_2_-NHCO-CCM-30, respectively. The irregular spherical shape is evident, and, despite agglomeration in both suspensions, identification of particles is possible.

Among the three functionalized nanoparticles, nSiO_2_-NHCO-CCM-30 was selected, hereafter referred to as nSiO_2_-NHCO-CCM, for the optical study, as it is the nanostructure that showed the highest degree of functionalization with curcuminoid molecules.

To verify the structure of the nanoparticles, samples of X-ray diffraction were analyzed from the powder of nSiO_2_, nSiO_2_-NHCO-CCM, and curcuminoid molecule **3**. Appendix A shows the diffractograms of nSiO_2_ powder, nSiO_2_-NHCO-CCM, and curcuminoid **3**. In the nSiO_2_ diffractogram, the amorphous nature of the nanoparticles was observed, with no diffraction peaks except for a broad band centered at 24°, characteristic of SiO_2_ nanoparticles [46]. The diffractogram of **3** exhibits sharp peaks between 10° and 30°, characteristic of a curcuminoid structure with a certain degree of amorphization [47]. In the case of nSiO_2_-NHCO-CCM, the broad signal of nSiO_2_ was observed along with a broad band coinciding with the most intense peak of **3**. This result indicates the presence of free curcuminoid that may be interacting through intermolecular forces with nSiO_2_-NHCO-CCM. However, a low-intensity broad band was observed around 40°, which could be attributed to the structural complexation of nSiO_2_ when chemically functionalized with the curcuminoid molecule.

### 3.3. UV–Vis Absorption Properties

The curcuminoid molecules **1**, **2**, and **3** exhibit a Donor–Acceptor–Donor electronic structure composed by the presence of donor groups (dimethylamine groups) at the aromatic ends of the π-conjugated system and an electron-acceptor keto-enol group in the central part. The structure of molecule 2 differs from that of molecule 1 solely due to the presence of an alkyl ester group (-(CH_2_)_2_COOCH_3_) in the α carbon between the keto-enol unit of the conjugated chain. As illustrated in Figure 4a, the optical properties of molecules **1** and **2** in CH_2_Cl_2_ remain very similar; they present absorption bands with a maximum at 488 nm and 500 nm, respectively, associated with a π-π* transition of the system in a keto-enol configuration where the entire molecule is fully conjugated [48]. This result indicates that the added aliphatic chain (-(CH_2_)_2_COOCH_3_) of molecule **2** minimally affects its electronic structure.

Previous observations indicated that the incorporation of electron-donor systems in curcuminoids promotes a red-shift of the absorption bands [49], but the -(CH_2_)_2_COOCH_3_ organic chain in molecule **2** can only provide electric density through inductive effects, exerting a minor influence on the shift of absorption bands. In addition, both molecules exhibit high and comparable absorption coefficients, with log(ε_max_) = 4.85 and 4.70 for **1** and **2**, respectively (Table 2), supporting the conclusion that the presence of the ester group at the α carbon insignificantly affects the absorption properties of the original molecule **1**. The slight difference between **1** and **2** represents a success for our strategy, where the -(CH_2_)_2_COOCH_3_ group was added exclusively to provide the molecule with a connection site to nSiO_2_ nanoparticles without altering the optical properties of **1**.

On the other hand, the maximum absorption band of compound **3** is located at 363 nm, showing dramatic shifts of more than 100 nm compared to its counterparts (Figure 4a). Nardo et al. demonstrated that curcumin containing a -CH_2_-COOH group at the α carbon between the keto-enol unit exhibited a more intense band at around 300 nm in various solvents. They argued that the -CH_2_-COOH group disrupts the resonance, originating from the movement of electrons through the aromatic rings, the heptadiene skeleton, and the ring structure in the keto-enol unit (as an intermolecular hydrogen bond), then, the band shifts to higher energies [9]. In comparison, it is reported that the optical properties of 4-(4-hydroxy-3-methoxyphenyl)-3-buten-2-one, popularly known as “half curcumin”, presents a single absorption band at 317 nm, attributed to reduced electronic mobility in the structure, requiring much more energy for the valence electrons to be promoted from the HOMO to LUMO orbitals [9,27]. These findings are consistent with the results obtained for molecule **3**, suggesting that the -CH_2_-COOH pendant group may reduce the complete conjugation of the molecular structure compared to compounds **1** and **2**. Nevertheless, the discrepancy in the position of the absorption band does not present a concern for our analysis, as molecule **3** primarily acts as an intermediate in the reaction, and the product of the reaction with an amide bond optically resembles molecule **2**.

In Figure 4a, the absorption spectrum of nSiO_2_-NHCO-CCM in a dichloromethane suspension is also presented. Despite successive washes of the reaction product, compound **3** persists in the nanoparticle, as evidenced by the peak around 368 nm, suggesting that, due to the high surface area of the nanoparticles, these molecules may be adsorbed and bound via intramolecular bonds. However, in addition to the absorption of **3**, a new signal was observed with a maximum of 472 nm, indicating the presence of the anchored curcuminoid molecule at a wavelength similar to that exhibited by compound **2**. This is supported by the fact that the carboxylic acid group of **3**, upon reacting with the -NH_2_ from the functionalized nanoparticle, converts into an amide bond with electronic behavior similar to an ester, as in molecule **2**. It is noteworthy that the band at 472 nm of nSiO_2_-NHCO-CCM was not observed in the bare nanoparticle (Appendix A), which exhibits an absorption band below 400 nm corresponding to defects in the nSiO_2_ structure [50,51], and it was not observed in curcuminoid **3** either, therefore providing a confirmation of the formation of an amide bond in the nSiO_2_-NHCO-CCM structure.

### 3.4. Fluorescence Properties

The emission spectra of curcuminoids **1**, **2**, and **3** were determined in dichloromethane solution (1.0 × 10^−5^ mol L^−1^) upon excitation at their absorption maxima. The fluorescence spectra are presented in Figure 4b, and the fluorescence data are detailed in Table 2. Small differences in the position of the emission maxima of **1** and **2** can be observed, with Stokes shifts of 77 nm and 73 nm, respectively. Compound **3**, with an emission band positioned at 441 nm, exhibits a Stokes shift of 76 nm. The absolute emission quantum yields (QY) of the three curcuminoids in dichloromethane were measured. From Table 2, it can be noted that the QY values for **1** and **2** were high and quite similar, while, for compound **3**, the molecule with the -(CH_2_)_2_COOH terminal carboxylate group, the QY value was below the detection limit of the equipment and could not be determined. Additionally, the fluorescence lifetimes (τ) of **1**, **2**, and **3** in a dichloromethane solution were studied using time-resolved fluorescence spectroscopy. Their fluorescence decays were fitted to a monoexponential function, and the data are presented in Table 2. Compounds **1** and **2** show short lifetimes of 1.45 and 1.32 ns, respectively, which can be explained by the presence of the methyl groups, which contain high-energy oscillator C\H groups in the conjugated system, leading to an increase in non-radiative decay [52]. Molecule **2** presents a lower lifetime than **1** due to the higher presence of C\H groups of the -(CH_2_)_2_COOCH_3_ pendant. On the other hand, curcuminoid **3** exhibits an even shorter time, which can be attributed to the complex formed by hydrogen bonds, which can lead to energy loss as heat [53]. In Figure 4b, the emission spectrum of nSiO_2_-NHCO-CCM from a suspension in dichloromethane is depicted at 538 nm, excited at 470 nm, distinct from the emission of the bare nanoparticle at the same excitation wavelength (Appendix A). This emission band centered at 538 nm closely resembles that exhibited by **2** at 573 nm, with Stokes shifts of 70 nm, which is within the same shifting range as molecules **1**, **2**, and **3**.

The relative emission quantum yield of the nanoparticles was determined using Rhodamine 6G as an actinometer [54,55]. The emission quantum yield and the fluorescence lifetime values are lower than those of the free molecules. Intermolecular interactions such as hydrogen bonding or van der Waals forces between portions of the nanoparticles and the curcuminoid molecule can generate fluorescence deactivation, as observed in other silicon oxide nanoparticles mixed with fluorophores [56]. However, due to the covalent incorporation of a curcuminoid molecule, it was possible to transfer optical properties such as absorption and emission in the visible range to the nanoparticle, a property which it did not initially possess.

### 3.5. Solvatochromic Effect

The solvent-induced spectral effects on the UV-Vis absorption and fluorescence behaviors of **1**, **2**, **3**, and nSiO_2_-NHCO-CCM were examined at room temperature (Figure 5). The UV-Vis absorption spectra of curcuminoids **1**, **2**, and **3** in thirteen different solvents are presented in Figure 5a for **1**, Figure 5b for **2**, Figure 5c for **3**, and in Figure 5d for nSiO_2_-NHCO-CCM. Variations in their photophysical properties depending on solvent polarity are detailed in Appendix A.

It was observed that the type of solvent has an effect on the UV-Vis absorption behavior of these compounds, **1**, **2**, **3** and nSiO_2_-NHCO-CCM. However, it can be observed that the greatest red-shift was found in dimethyl sulfoxide, while the smallest one was observed in hexane, for **1**, **2**, and **3**, indicating a bathochromic shift as solvent polarity increases. In the case of nSiO_2_-NHCO-CCM, the effect is subtle and highlights the presence of dispersive effects (broad bands) between the nanoparticles and the surrounding medium, limiting absorbance analysis.

For a more detailed account of the solvent-induced spectral effects, DFT and TD-DFT calculations were performed for six solvents of contrasting polarity (see Section 2 for further technical detail). Hexane, benzene, chloroform, acetone, acetonitrile, and dimethyl sulfoxide were chosen with the purpose of demonstrating the existence and theoretical manifestation of the bathochromic effect. In the case of **1** and **2**, the calculated UV-Vis spectra were in close agreement with the experimental behavior since a clear bathochromic shift was observed as the solvent polarity was increased (see Figure 6a,b). To check whether the electronic communication associated with the bonding to the SiO_2_ nanoparticle can affect the observed bathochromic shift, we constructed the model depicted in Figure 6c. TD-DFT spectra indicate a similar behavior to **1** and **2** and nSiO_2_-NHCO-CCM, suggesting that the optical properties of CCM can be retained after functionalization.

As can be seen in Figure 7, the compounds display a pronounced positive solvatofluorochromism that is notably more intense and significant compared to the absorption spectroscopy results. It was observed that their fluorescence maxima underwent a marked bathochromic shift from the least-polar solvent (hexane) to the most polar (dimethylsulfoxide), in terms of its dielectric constant, showing differences of 486 to 606 nm, 502 to 601 nm, 390 to 451 nm, and 517 to 540 nm for **1**, **2**, **3**, and nSiO_2_-NHCO-CCM, respectively (Appendix A). These results suggest that the enhancement in solvent polarity exerts a more pronounced influence on the energy of the excited state as opposed to the ground state [48]. This observation is justified by the stabilization of the excited state more than that of the ground state, which reduces the energy difference between the ground and excited states, resulting in a red-shift of the corresponding band.

It is evident that the red-shift in more-polar solvents is discrete in the case of nSiO_2_-NHCO-CCM nanoparticles. Additionally, it was observed that, in hydrogen bond acceptor solvents such as ethanol, acetonitrile, acetone, and ethyl acetate, an additional shoulder to the more intense signal is present. It is known that curcuminoid molecules exhibit broad emission bands due to the presence of different species in equilibrium, and this diversity can be even greater when the nanoparticle is part of the emitting system. However, these species might be stabilized by parameters other than solvent polarity. To study the effect of various solvent parameters on the spectral shift of the emission bands, a multilinear regression analysis (Linear Solvation Energy Relationships (LSER)) was carried out based on the Kamlet–Taft (1) and Catalán (2) solvent scales [57,58]. In this way, information is obtained on how different parameters defined in these scales affect the spectral shift of species **1**, **2**, **3**, and nSiO_2_-NHCO-CCM (see Section 2 for further technical detail).

Through the Kamlet–Taft and Catalán solvent scales, it is possible to discern the influence of non-electrostatic interactions versus electrostatic interactions. Non-electrostatic solute–solvent interactions, such as hydrogen bonding, are expressed through the solvent acidity, SA or α, and solvent basicity, SB or β. On the other hand, electrostatic interactions are described by the π* parameter in the Kamlet–Taft equation [38], encompassing both polarizability and dipolarity. In contrast, in the Catalán equation [41], these parameters are expressed independently as SP and SdP. This separation represents an advantage over the Kamlet–Taft approach, as it allows for the individual assessment of the effect of solvent dipolarity and polarizability on the spectral shift of the analyzed species. In both solvent scales, the Kamlet–Taft and Catalán, ν_o_ is the regression value of the solute property in the reference solvent. The remaining parameters (coefficients a, b, s, c, and d) are obtained through multilinear regression analysis and estimate the relative contribution of solvent molecules in the photophysical behavior of solute molecules. In Appendix A, the physical properties and parameters of all solvents used in this work are shown, while Table 3 and Table 4 present the results of the multilinear regression analysis adjusted with a 95% confidence level for both equations.

In Table 3, it can be observed that the a, b, and c parameters the Kamlet–Taft equation for compounds **1**, **2**, and **3** exhibit negative values, indicating that all three parameters contribute to a positive solvatofluorochromism. It is noteworthy that parameter *s* has the most negative value in all three cases. This suggests that the solvent’s dipolarity/polarizability is the factor causing the most significant changes in the emission frequency of compounds **1**, **2**, and **3**, resulting in a more pronounced red-shift in the emission band. Similarly, in the case of nSiO_2_-NHCO-CCM, this trend persists, where parameter *s* has the highest absolute value among the three parameters but is of a smaller magnitude compared to **1**, **2**, and **3**. This implies that the trend of the red-shift in the emission band with increasing dipolarity/polarity of the solvent continues, albeit to a lesser extent than observed in the free curcuminoid molecules. On the other hand, Table 4 shows the results of the Catalán parameters. A trend can be observed in the values of the a, b, c, and d parameters for **1**, **2**, and **3**. In most cases, the parameters follow the order in absolute value of a~d > c > b. This suggests that the solvent’s capacity to establish hydrogen bonds and its polarizability plays a substantial role in the spectral emission shift. The solvent’s ability to donate hydrogen bonds has a more pronounced effect on the transition energy than its ability to accept hydrogen bonds. In the Catalán equation, it is possible to conduct an independent analysis of the influence of solvent dipolarity and polarizability on the emission band position. In this context, the solvent’s polarizability exerts a more prominent influence than dipolarity, an analysis that is not feasible through the Kamlet–Taft equations, as the π* parameter involves both concepts. Regarding nSiO_2_-NHCO-CCM, it was observed that the parameter with the greatest impact on the emission band position, according to the Kamlet–Taft equation, is a (π*), while, for the Catalán equation, it is d, representing solvent polarizability. By using both equations, the same result is obtained; as the solvent becomes more dipolar, a pronounced bathochromic shift in the emission band prevails. However, the values of a and d are smaller than those exhibited by **1**, **2**, and **3**, implying that the effect of polarizability is present but more discreet than in free curcuminoid molecules.

Table 5 and Table 6 illustrate the percentage contributions to the parameter values for Equations (1) and (2). In the case of Kamlet–Taft equation, the π* parameter is predominant, while, for Catalán equation, the ability to form hydrogen bonds and polarizability contribute equally significantly to the spectral shift. These results also suggest the presence of an emissive state characterized by significant intramolecular charge transfer. Therefore, a greater stability of the first excited state is observed in the presence of a highly polarizable solvent compared to the ground state. This characteristic was transferred to the nSiO_2_-NHCO-CCM system, indicating that the nanoscale system affects the optical properties inherent to free curcuminoid discreetly.

To validate the robustness of the multilinear regression between the Catalán and Kamlet–Taft equations, a graphical comparison was conducted between the experimental emission frequencies (ν_o,exp_) and the calculated ones (ν_o,cal_). The linearity of this function is closely related to a strong correlation, indicating that, as the value of R (correlation coefficient) increases, the model fits the experimental data better. As can be seen in Appendix A, a correlation coefficient fit exceeding 0.90 is achieved for all evaluated systems.

Therefore, the analysis of solvent effects reveals the significant influence of polarizability/polarity on the spectral properties of both free curcuminoids and nSiO_2_-NHCO-CCM nanoprobes. It is noteworthy that the bathochromic shift trends observed in highly polar solvents are also evident in the manufactured nanosystem, as confirmed by experimental, computational, and multilinear regression results. These findings underscore the importance of understanding the role of solvent in modulating the optical properties of the compounds under study, which could have significant implications in areas such as sensing and organic liquid detection.

### 3.6. Nanosensor Performance for Detecting Organic Liquids

A preliminary evaluation of the nSiO_2_-NHCO-CCM hybrid nanosystem was conducted to detect kerosene, an organic liquid commonly associated with contamination incidents [59]. Two approaches were employed: firstly, we examined its behavior in the presence of 100% kerosene, and, secondly, we evaluated its detection capability in water containing kerosene with a 1 µL/mL concentration. In both cases, the hybrid nanoparticles had a concentration of 0.3 mg/mL in water. These results show that kerosene alone emits discreetly, an emission attributed to the hydrocarbons present in its composition. This phenomenon is amplified in the presence of the hybrid nanoparticles, as observed in Figure 8a, which shows a signal with a maximum around 525 nm. In Figure 8b, it is evident that, in aqueous solutions with low concentrations of kerosene, the hybrid nanoparticles can provide a fluorescent emission response too. The fact that the signal is activated demonstrates its potential in detecting organic liquid contaminants in water. It is important to note that, in water, the hybrid nanoparticles do not become deactivated, unlike the free curcuminoid molecule, which is not soluble in water. However, future work should explore the sensitivity to different concentrations and mixtures of this contaminant and others in water.

## 4. Conclusions

The successful syntheses of a set of three curcuminoid molecules (**1**, **2**, and **3**) with terminal dimethylamine groups were presented. These curcuminoid molecules with terminal dimethylamine groups showed a significant increase in quantum yield, reaching values around 0.60 in dichloromethane (curcuminoids **1** and **2**), well above what has been reported for curcumin. Additionally, a functionalization strategy was implemented by adding an aliphatic chain with a terminal carboxylic acid group to the central carbon of the curcuminoid structure, enabling successful binding to silica oxide nanoparticles. The characterization of molecules **1**, **2**, and **3** was carried out using various analytical techniques such as FTIR-ATR, ^1^H-RMN, UV-Vis and fluorescent emission spectroscopy, and MALDI-TOF. Once the curcuminoid molecules were synthesized and characterized, curcuminoid **3** was anchored onto nSiO_2_ nanoparticles by the formation of amide bonds. This process involved the reaction between the -NH_2_ terminal groups of functionalized nSiO_2_ nanoparticles and the carboxylic acid terminal group of a central chain in the curcuminoid molecule **3**. Characterization of the hybrid nSiO_2_-NHCO-CCM nanoparticles confirmed the successful functionalization, and micrographs revealed that the nanoparticles retained their size and morphology after functionalization. The bare nSiO_2_ nanoparticles are stable in solution but lack optical properties. After functionalization with the curcuminoid molecule, the nSiO_2_-NHCO-CCM nanoparticles acquired the optical properties of the curcuminoid molecule anchored to the system. In terms of optical properties, the nSiO_2_-NHCO-CCM nanoparticles exhibited absorption and emission bands with maxima at 472 and 538 nm, respectively, in dichloromethane, while the original curcuminoid molecule 1 showed maxima values at 448 and 563 nm. Furthermore, the multilinear regression analysis (Linear Solvation Energy Relationships (LSER)) analysis using the Kamlet–Taft and Catalán equations revealed that both the first excited state of free curcuminoid molecules and that of nSiO_2_-NHCO-CCM nanoparticles follow the same pattern, stabilizing mainly in dipolar solvents. This is reflected in the more negative values of π and SP, both negative and of greater magnitude, resulting in red-shifts in the emission band compared to solvents with low polarizability. Preliminary evaluation of the hybrid nanosystem in the detection of kerosene has demonstrated its ability to detect this contaminant through fluorescent emission, suggesting its potential utility for detecting organic liquids in water. However, further research is needed to improve the sensitivity and specificity of the detection.

## Figures and Tables

**Figure 1 nanomaterials-14-01022-f001:**
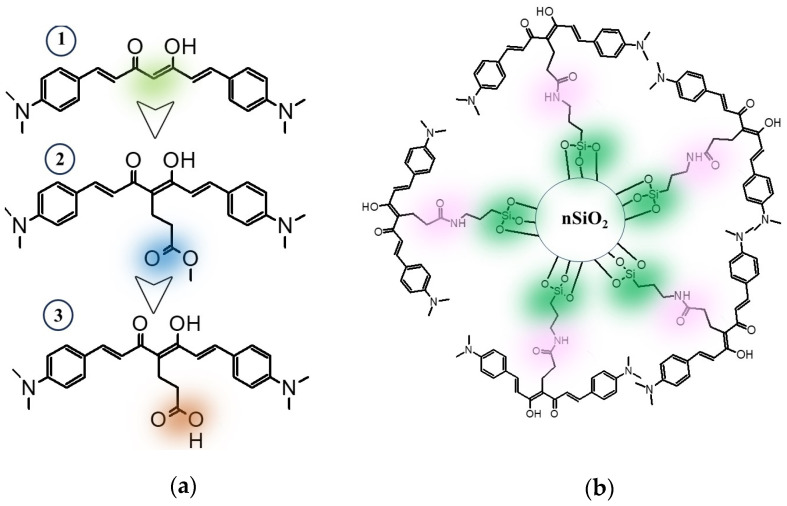
(**a**) Molecular structures of synthesized curcuminoid molecules. (**b**) Drawing representing the anchored curcuminoid on the surface of silicon oxide nanoparticles.

**Figure 2 nanomaterials-14-01022-f002:**
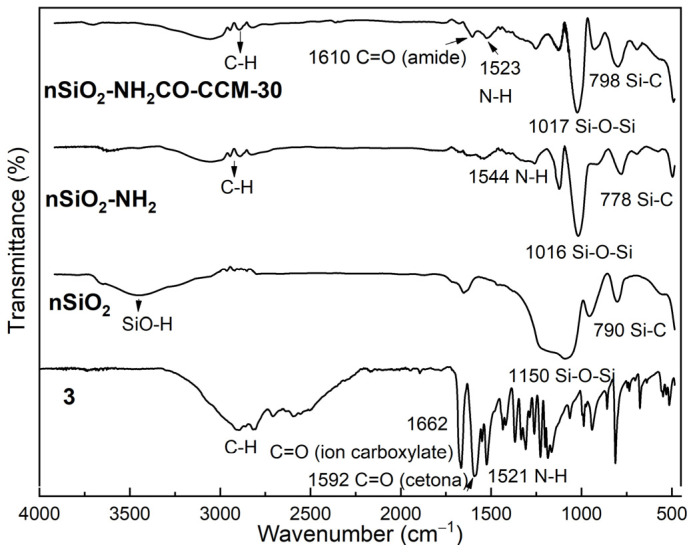
FTIR-ATR spectra of molecule **3**, bare nanoparticles (nSiO_2_), nanoparticles functionalized with APTES (nSiO_2_-NH_2_), and nanoparticles anchored to curcuminoid.

**Figure 3 nanomaterials-14-01022-f003:**
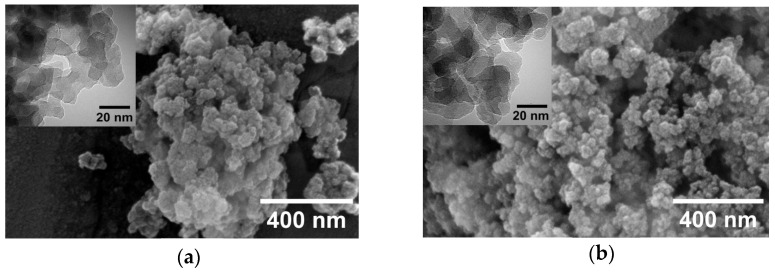
FESEM micrograph of (**a**) nSiO_2_ and (**b**) nSiO_2_-NHCO-CCM-30. The insets in (**a**) and (**b**) correspond to TEM micrographs of nSiO_2_ and nSiO_2_-NHCO-CCM-30, respectively.

**Figure 4 nanomaterials-14-01022-f004:**
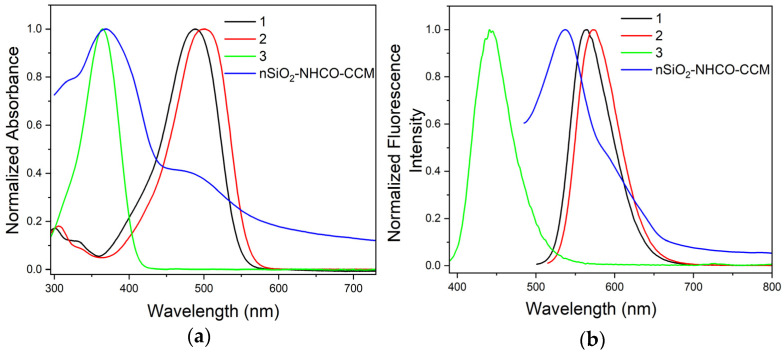
Absorption (**a**) and emission spectra (**b**) of curcuminoid molecules **1**–**3** and nSiO_2_-NHCO-CCM suspension in CH_2_Cl_2_.

**Figure 5 nanomaterials-14-01022-f005:**
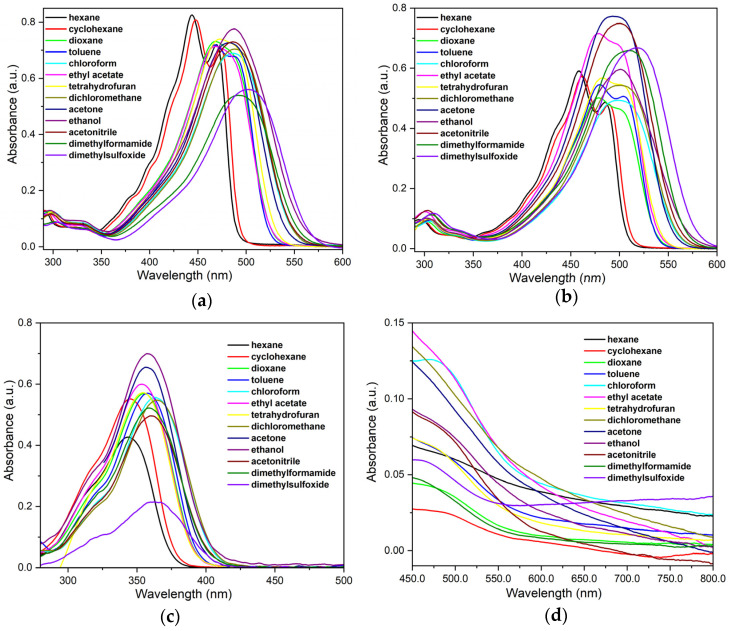
Absorption spectra of (**a**) curcuminoid 1, (**b**) curcuminoid 2, (**c**) curcuminoid 3, and (**d**) nanosystem in thirteen solvents.

**Figure 6 nanomaterials-14-01022-f006:**
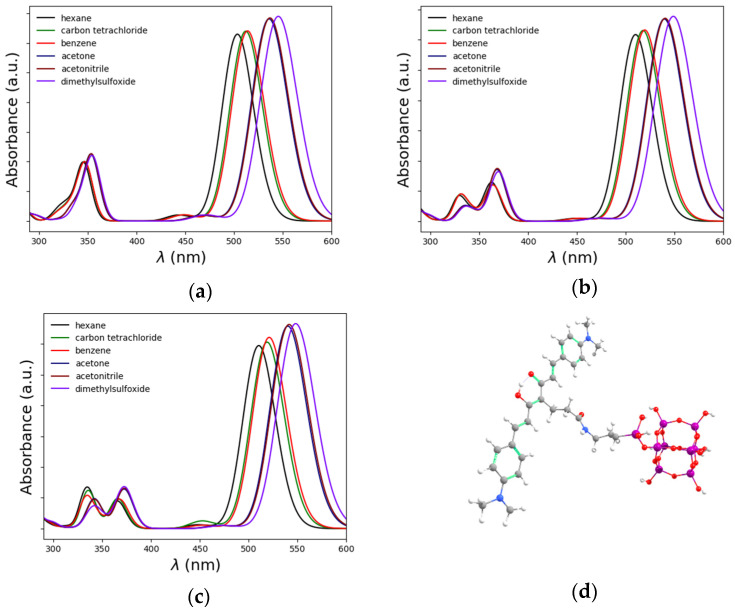
TD-DFT calculated absorption spectra of 1 (**a**), 2 (**b**), nSiO_2_-NHCO-CCM and the optimized model of CCM bonded to an SiO_2_ unit (**c**), Depicted model of the SiO_2_ nanoparticle functionalized with the curcuminoid molecule (**d**), Color code: Si (purple), O (red), N (blue), C (gray) and H (white).

**Figure 7 nanomaterials-14-01022-f007:**
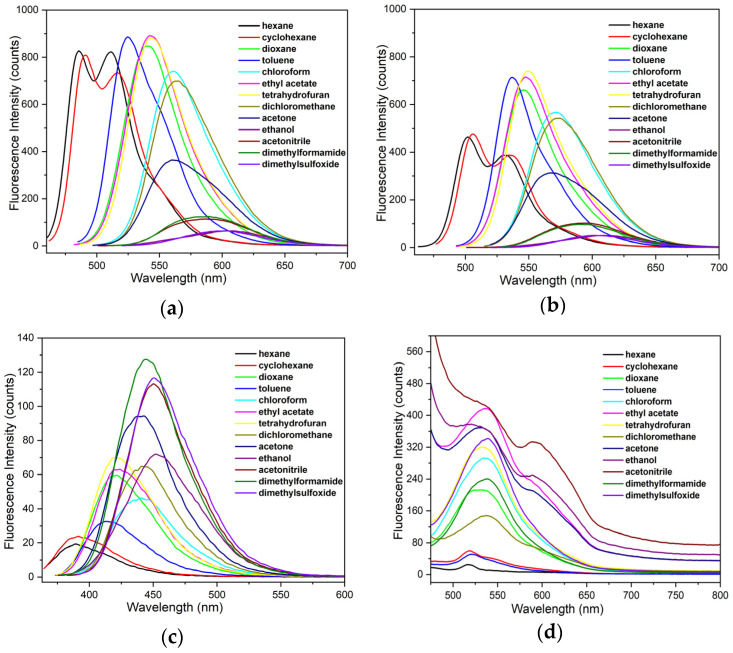
Emission spectra of (**a**) curcuminoid **1**, (**b**) curcuminoid **2**, (**c**) curcuminoid **3**, and (**d**) nSiO_2_-NHCO-CCM nanosystem in thirteen solvents. For the emission spectra, **1**, **2**, and **3** were excited at their absorption maxima and nSiO_2_-NHCO-CCM at 470 nm.

**Figure 8 nanomaterials-14-01022-f008:**
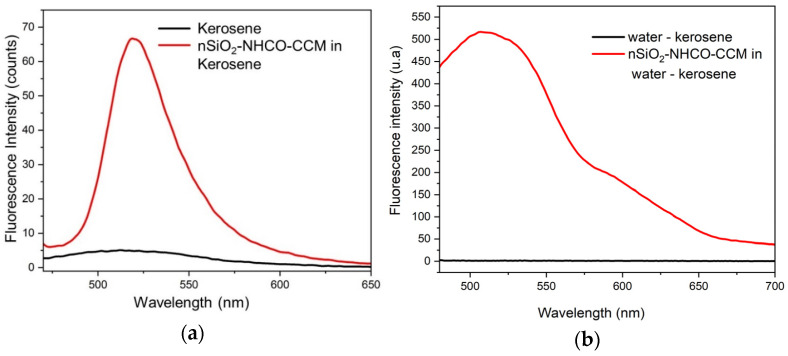
Spectral emissions: (**a**) kerosene and a suspension of nSiO_2_-NHCO-CCM in kerosene, (**b**) a suspension of nSiO_2_-NHCO-CCM in water with kerosene.

**Table 1 nanomaterials-14-01022-t001:** Values obtained through the Kaiser colorimeter assay for -NH_2_-functionalized nanoparticles and the nSiO_2_-NHCO-CCM obtained using the three different reaction stoichiometries.

	Absorbance at 579 nm	Amino Group Concentration (μmol/g Nanoparticle)	Degree of Functionalization of Curcuminoid Molecule (μmol/g Nanoparticle)	Z Potential (mV)
nSiO_2_-NH_2_	0.5790 ± 0.0361	238 ± 17.2	-	−27.5
nSiO_2_-NHCO-CCM-10	0.1385 ± 0.0153	121 ± 12.1	118	37.8
nSiO_2_-NHCO-CCM-30	0.0022 ± 0.0014	17 ± 5.8	222	40.9
nSiO_2_-NHCO-CCM-50	0.0045 ± 0.0009	29 ± 5.9	210	41.3

**Table 2 nanomaterials-14-01022-t002:** Photophysical data of curcuminoids **1**–**3** and nSiO_2_-NHCO-CCM in CH_2_Cl_2_ solution.

Compound	λ_abs_ (nm)	Log(ε_max_)	λ_em_ (nm)	τ (ns)	Φ_em_
**1**	488	4.85	565	1.45	0.69 ^a^
**2**	500	4.74	573	1.32	0.63 ^a^
**3**	363	4.74	441	0.75	-
nSiO_2_-NHCO-CCM	472	-	538	0.13	0.017 ^b^

^a^: Measurement of absolute quantum yield. ^b^: Measurement of relative quantum yield using Rhodamine 6G as actinometer.

**Table 3 nanomaterials-14-01022-t003:** Regression analysis applied to the solvatochromic parameters of Kamlet–Taft equation.

Compounds	1	Standard Error	t-Statistic	P(2-tail)
υ_o_	20,385	223	91	<0.0001
s	−3170	417	−8	<0.0001
a	−767	447	−2	0.12019
b	−1758	467	−3	0.004
Coefficient of multiple correlation = 0.969			F = 46	N = 13
**Compounds**	**2**	**Standard Error**	**t-Statistic**	**P(2-tail)**
υ_o_	19,773	225	88	<0.0001
s	−2753	420	−7	<0.0001
a	−400	450	−1	0.3976
b	−1825	470	−4	0.0037
Coefficient of multiple correlation = 0.958			F = 33	N = 13
**Compounds**	**3**	**Standard Error**	**t-Statistic**	**P(2-tail)**
υ_o_	25,453	265	96	<0.0001
s	−3369	495	−7	<0.0001
a	−55	530	−0.1	0.9199
b	−1672	554	−3	0.0145
Coefficient of multiple correlation = 0.951			F = 28	N = 13
**Compounds**	**nSiO_2_-NHCO-CCM**	**Standard Error**	**t-Statistic**	**P(2-tail)**
υ_o_	1925	82	234	<0.0001
s	−810	53	−5	<0.0001
a	92	16	−0.6	0.5892
b	418	17	−2	0.0380
Coefficient of multiple correlation = 0.910			F = 14	N = 13

**Table 4 nanomaterials-14-01022-t004:** Regression analysis applied to the solvatochromic parameters of Catalán equation.

Compounds	1	Standard Error	t-Statistic	P(2-tail)
υ_o_	22,754	862	26	<0.0001
a	−3722	818	−5	0.0018
b	−242	444	−0.5	0.6009
c	−2826	302	−9	<0.0001
d	−3716	123	−3	0.0016
	Coefficient of multiple correlation = 0.985		F = 65	N = 13
**Compounds**	**2**	**Standard Error**	**t-Statistic**	**P(2-tail)**
υ_o_	21,456	756	28	<0.0001
a	−3804	717	−5	<0.0001
b	322	389	−0.8	0.4321
c	−2604	265	10	<0.0001
d	−2719	107	−2	0.0356
	Coefficient of multiple correlation = 0.958		F = 63	N = 13
**Compounds**	**3**	**Standard Error**	**t-Statistic**	**P(2-tail)**
υ_o_	26,758	1090	24	<0.0001
a	−2801	1034	−2.7	0.0267
b	630	561	−1.1	0.2934
c	−3185	382	−8	<0.0001
d	−2225	155	−1.4	0.0190
	Coefficient of multiple correlation = 0.973		F = 35	N = 13
**Compounds**	**nSiO_2_-NHCO-CCM**	**Standard Error**	**t-Statistic**	**P(2-tail)**
υ_o_	20,434	437	47	<0.0001
a	1063	415	−2.5	0.0336
b	−243	225	−1.1	0.3114
c	−416	15	−2.7	0.0266
d	−1797	624	−2.9	0.0205
	Coefficient of multiple correlation = 0.912		F = 10	N = 13

**Table 5 nanomaterials-14-01022-t005:** Percentage contribution of solvatofluorochromic parameters (Equation (1)).

Compounds	P_π*_ (%)	P_α_ (%)	P_β_ (%)
**1**	55.7	13.5	30.9
**2**	55.3	8.0	36.7
**3**	66.1	1.1	32.8
nSiO_2_-NHCO-CCM	61.4	7.0	31.7

**Table 6 nanomaterials-14-01022-t006:** Percentage contribution of solvatofluorochromic parameters (Equation (2)).

Compounds	P_SA_	P_SB_	P_SdP_	P_SP_
**1**	35.4	2.3	26.9	35.4
**2**	40.3	3.4	27.6	28.8
**3**	31.7	7.1	36.0	25.2
nSiO_2_-NHCO-CCM	30.2	6.9	11.8	51.1

## Data Availability

Data are contained within the article and Appendix A.

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
