# Peer review of "Synthesis and Optical Properties of a Novel Hybrid Nanosystem Based on Covalently Modified nSiO2 Nanoparticles with a Curcuminoid Molecule"

_nanomaterials, 2024, doi:10.3390/nano14121022_

Round 1

Reviewer 1 Report

Comments and Suggestions for Authors

This is a manuscript about the preparation of silica nanoparticles with covalently-bound curcuminoid molecules serving as probe for the detection of kerosene by fluorescence response. The manuscript has merit, however there are several issues and queries that must be addressed first, ranging from minor to major:

Line 34: Photophysical (not "Photophysics")

Line 38: rephrase to "has its origins"

Line 125: purchased (past tense)

Lines 145, 192, 237, 256-258, 339, 354: use proper citing style of this journal (with square brackets, not superscripts). Numbers should correspond to the order of references in the Reference List and numbering should be in the sequence of referral in the text

Lines 164 and 172: There appears to be a systematic positive error in masses determined by MALDI-TOF. Has the equipment been properly calibrated before the experiments?

Line 278, 283 etc. : Supplementary figures should also be referred in their normal sequence in the main text of the paper. For example, you mention Fig. S1 and S3 after Fig. S4 which has been already mentioned on line 228

Line 295: I believe you meant to say "previously", not "preciously"

Table 1 column heading: nanoparticle (spelling!)

Table 1: since standard deviation values are shown, it means replication of experiments has been performed, but you do not mention how many replications were done.

Figure 2: Why are the two expected to be prominent NH2 stretches (symmetric and asymmetric) not visible in the 3300-3500 reciprocal centimeters range of the IR spectrum of nSiO2-NH2?

Figure 5 legend is inconsistent: only Dimethylformamide is capitalized. Please be consistent in presentation/formatting.

Tables 3 and 4: some advanced statistics tests appear to have been performed on the date according to these tables, but nowhere are the details of these statistic tests described. Include such a section in the Materials & Methods part.

The part about the application of the nanosensor for determining kerosene is very incipient as described. More investigations should be done, regarding the linear range, LOD, LOQ, interferences, sensitivity of the device and method.

References are not formatted in the style demanded by this journal!

Comments on the Quality of English Language

English is fine for the most part, a few minor issues here and there that will be addressed easily by the authors themselves during revisions

Author Response

Comments and questions of the reviewers an3d our responses:

Reviewer 1: This is a manuscript about the preparation of silica nanoparticles with covalently-bound curcuminoid molecules serving as probe for the detection of kerosene by fluorescence response. The manuscript has merit, however there are several issues and queries that must be addressed first, ranging from minor to major:

Answer: Thank you to Reviewer 1 for considering our work a contribution to the knowledge of the field of functionalization of materials. We modified the manuscript according to his comments and suggestions. Each change related to Reviewer 1's comments is written in color font in the new version of our manuscript. Below, we answer each of your comments in detail.

Line 34: Photophysical (not "Photophysics") 

R = We thank the reviewer 1 for this remark. It has been corrected

Line 38: rephrase to "has its origins"             

R = We thank the reviewer 1 for this remark. It has been corrected

Line 125: purchased (past tense)        

R = We thank the reviewer 1 for this remark. It has been corrected

Lines 145, 192, 237, 256-258, 339, 354: use proper citing style of this journal (with square brackets, not superscripts). Numbers should correspond to the order of references in the Reference List and numbering should be in the sequence of referral in the text

R = We thank the reviewer 1 for this remark. We changed the citing style with square brackets, no superscripts. And we organized the numbers to correspond to the order of the references in the text.

Lines 164 and 172: There appears to be a systematic positive error in masses determined by MALDI-TOF. Has the equipment been properly calibrated before the experiments?

R = We thank Reviewer 1 for their observation. In response, the nomenclature for the calculated data was modified. For example, the calculated value for curcuminoid 1 is now specified as [C23H26N2O2 + H]+: 363.46 and the found value as 363.32. With this modification, we believe that the characterization results are more clearly understood. Additionally, answering the question the MALDI spectra were obtained using a MALDI-TOF Microflex instrument (Bruker Daltonics Inc., MA, USA) with α-cyano-4-hydroxycinnamic acid and 2,5-dihydroxybenzoic acid as matrices. Prior to acquiring the spectra, the instrument was calibrated with an external standard in the m/z range of 280-1200. The spectrometer was controlled using the flexControl 3.0 software. For the detection of monoisotopic m/z signals, the MALDI-TOF Peptides algorithm with default parameters was employed instrument's accuracy. We have included this information in the methodology section for clarity regarding the measurements.

Line 278, 283 etc. : Supplementary figures should also be referred in their normal sequence in the main text of the paper. For example, you mention Fig. S1 and S3 after Fig. S4 which has been already mentioned on line 228

R = We thank the reviewer 1 for this observation. We have organized the text so that the figures in the manuscript and the supplementary document are in correlative order.

Line 295: I believe you meant to say "previously", not "preciously"

R = We thank the reviewer 1 for this remark. It has been corrected.

Table 1 column heading: nanoparticle (spelling!)

R = We thank Reviewer 1 for this remark. It has been corrected.

Table 1: since standard deviation values are shown, it means replication of experiments has been performed, but you do not mention how many replications were done.

R = Thanks to Reviewer 1 for this comment. We have added in the manuscript in the Methods section, that the absorbance spectra measurements for constructing the calibration curve, as well as the measurements of the nanoparticles with covalently-bound curcuminoid molecules, were performed in triplicate.

Figure 2: Why are the two expected to be prominent NH2 stretches (symmetric and asymmetric) not visible in the 3300-3500 reciprocal centimeters range of the IR spectrum of nSiO2-NH2?

R = We thank Reviewer 1 for this observation. In response to your comment, the infrared spectrum in the wavenumber range of 3300-3500 cm⁻¹ should exhibit the stretching of N-H and O-H bonds, both present in nSiO2-NH2. The signal at 3300-3500 cm⁻¹ for nSiO2-NH2 has decreased in intensity compared to nSiO-H because many of the O-H groups on the nanoparticle have reacted to form the Si-O bond with 3-aminopropyltriethoxysilane. However, the functionalization is not completed, leaving some nSiO-H groups present. These groups form hydrogen bonds, resulting in broad bands around 3300-3500 cm⁻¹, which makes the N-H stretching difficult to be observed clearly.

Figure 5 legend is inconsistent: only Dimethylformamide is capitalized. Please be consistent in presentation/formatting.

R = We thank the Reviewer 1 for this comment. Figure legends were modified following the Reviewer's suggestions.

Tables 3 and 4: some advanced statistics tests appear to have been performed on the date according to these tables, but nowhere are the details of these statistic tests described. Include such a section in the Materials & Methods part.

R = We thank Reviewer 1 for this remark, which has helped us to improve our manuscript significantly. We have included a detailed description of the methodology used for the multilinear regression analysis.

The part about the application of the nanosensor for determining kerosene is very incipient as described. More investigations should be done, regarding the linear range, LOD, LOQ, interferences, sensitivity of the device and method.

R = We thank Reviewer 1 for this observation. The purpose of this study is to highlight the innovative covalent modification of nSiO2 nanoparticles with a curcuminoid molecule. The detection assay serves as a preliminary test to demonstrate its potential as a detection probe. However, the ultimate goal is to exhibit the covalent modification of the nanoparticle. This strategy starts with the design of a new curcuminoid molecule for anchoring covalently to the nanoparticles treated with APTES molecules to provide amino groups for covalent bond formation. We apologize for any confusion. To clarify the focus of this work, we have revised the manuscript, highlighting in color the words we have modified for this purpose.

References are not formatted in the style demanded by this journal!

R = We thank Reviewer 1 for this remark. It has been corrected.

Comments on the Quality of English Language

English is fine for the most part, a few minor issues here and there that will be addressed easily by the authors themselves during revisions

Reviewer 2 Report

Comments and Suggestions for Authors

attached

Comments on the Quality of English Language

nil

Author Response

Reviewer 2:

Answer: Thanks to Reviewer 2 for the feedback and suggestions. We have rewritten part of the manuscript to put more emphasis on the novelty and relevance of our work in the field of chemical functionalization. Each change related to Reviewer 2's comments is highlighted in color in the new version of the manuscript.

Line 29: The Abstract says the emission is around 516nm and the results say around 525nm. Which is correct?

R = We thank the Reviewer 2 for this observation and agree with it. We have corrected the value in the abstract, and the correct value is 525 nm.

Line 30: There is no result for nSiO2-NH-CCM in the article.

R = We thank Reviewer 2 for this observation. . There was a mistake. We have change the nomenclature to the correct form nSiO2-NHCO-CCM

Line 108: In Figure 1b why 2 groups of SiO2 are empty and why all SiO2 groups are not identical as some contain H and some do not.

R = We thank Reviewer 2 for this observation. Indeed, we have modified Figure 1b to ensure that all connections to the nanoparticle are identical.

Line 156: Whichboron complex was formed and how?

R = We thank Reviewer 2 for this observation. We have added a synthesis scheme (Scheme S1) that shows an intermediate in the reaction to obtain curcuminoid molecules is a boron complex. This complex is formed between a boron ion and the units of either acetylacetone or methyl-4-acetyl-5-oxohexanoate, which, in the final step, decomposes to generate the free curcuminoid molecules.

Line 178: The chemical name that was used to maintain pH was not mentioned.

R = We thank Reviewer 2 for this remark. We have used 10% HCl to achieve the desired pH, and we have added this information to the manuscript.

Line 195: The symbol used for micro is not correct and in other places check the symbol of micro.

R = We thank Reviewer 2 for this observation and agree with it. We have corrected the symbol of micro.  

Line 233: Explain what is pedant acid

R = We thank Reviewer 2 for this remark. In the manuscript, we have replaced the term "pendant acid" with "a central -(CH2)2-COOH chain" or "central chain" We believe this is a clearer way to refer to the functional group that enables covalent bonding with the nanoparticles.

Line 327: Fig. 2 Unit of transmittance is % not a.u

R = We appreciate the Reviewer's 2 comments. We have corrected the units for all infrared spectra in the manuscript and the supplementary material.

Line 411:  Fig 4 Should provide UV spectra of nSiO2 for comparison

R = We thank Reviewer 2 for this comment, which helped us improve our discussion and reinforce the confirmation of the covalent functionalization of the SiO2 nanoparticle. We have added Figure S9 in the supplementary material, where the absorption and emission spectra of the bare nanoparticle and the nanoparticle functionalized with the curcuminoid molecule overlap. In this image, it can be observed that in the absorption region of nSiO2-NHCO-CCM, the bare nanoparticle does not exhibit absorption. On the other hand, in the emission spectrum, it is evident that under the same measurement conditions, the bare nanoparticle presents a strong noise, which supports its low or null emission capacity. We decided to generate a new Figure S9 instead of directly adding the spectra of the bare nanoparticle to Figures 4a-b, considering that these are normalized graphs and the inclusion of the nSiO2 data could reduce the clarity of Figure 4.

Line 617: No "c" parameter is present in Table 3

R = We thank Reviewer 2 for this comment. To clarify, in the Kamlet-Taft equation (Equation 1, Table 3), the parameters "s," "a," and "b" were defined, while the parameter "c" was defined for the Catalan equation (Equation 2). Therefore, the parameter "c" is only found in Table 4, which refers to the coefficients of the Catalan equation, while Table 3 refers to the coefficients of the Kamlet-Taft equation.

Line 516: Fig. 5 No absorbance unit is mentioned and no absorbance value is mentioned

R = We thank Reviewer 2 for this remark. In response to their comment, we have made the necessary adjustments to the absorbance spectra in the manuscript and have included an explicit mention in the text regarding the location of the absorption, emission, and Stokes shift values for 1, 2, 3, and nSiO2-NHCO-CCM in Table S2. Additionally, we would like to inform them that in the manuscript text, we have focused on highlighting the most relevant trends, while specific absorption and emission values have been detailed and analyzed alongside Figure 4, which is presented in dichloromethane.

Line 543: Fig. 6 No absorbance unit is mentioned

R = We thank Reviewer 2 for this remark. It has been corrected.

Line 718: Statement not comply with figure 4

R = We thank Reviewer 2 for this remark. In Figure 4, we have indeed limited our analysis to a single solvent. For this reason, we have chosen to address the study of the redshift of the bands as a function of solvent polarity increase based on the results presented in Figure 5. This strategy allows us to examine this key aspect more thoroughly and provide a comprehensive analysis of our research.

Suggestions from Reviewer 2:

  1. Improve the language of the article.

R = We appreciate Reviewer 2 for this comment. We have improved the language of the article, taking into consideration the feedback provided by both Reviewer 1 and Reviewer 2.

  1. Should do XRD of all materials.

R = We appreciate the Reviewer 2 suggestion. We have added Figure S8 to the supplementary material, illustrating the X-ray diffraction patterns of the bare nanoparticle, the nanoparticle functionalized with the curcuminoid molecule, and curcuminoid molecule 3. Additionally, we have included a paragraph analyzing the diffraction patterns of these three species, highlighting the differences and similarities among them.

  1. Should give synthesis scheme of prepared materials

R = We value the suggestion provided by the Reviewer 2. In response, we have decided to include schematic schemes depicting the synthesis of the curcuminoid molecule and the functionalization of the nanoparticle. These illustrations can be found in the supplementary material accompanying this study.

  1. Give pictures of prepared materials.

R = We appreciate the Reviewer 2 suggestion. We have included the images in the supplementary material as insets in each spectrum, shown in Figure S8.

Comments from Reviewer 2:

  1. The morphology of the silica nanoparticles was irregular, which could affect reproducibility.

R = We thank Reviewer 2 for this comment. We want to highlight that three different functionalizations were carried out: nSiO2-NHCO-CCM-10, nSiO2-NHCO-CCM-30, and nSiO2-NHCO-CCM-50. In all cases, there is evidence of functionalization. Using the Kaiser test, it was possible to determine the degree of functionalization for each sample, and in the infrared spectrum, a band attributable to the new C=O (amide) bond is observed. All these functionalized nanoparticles exhibit a Z potential and a hydrodynamic diameter distinct from the bare nanoparticle. This consistency suggests that, despite the irregular morphology of the silica nanoparticles, the covalent bonding of the curcuminoid was effectively achieved under all tested functionalization conditions. This corroborates the fact that the morphological irregularity of the nanoparticles does not affect the reproducibility of the process.

  1. Why were only 15-20nm SiO2 nanoparticles chosen for this study? And does size affect functionalization?

R = We thank Reviewer 2 for this comment. We selected SiO2 nanoparticles ranging from 15 to 20 nm primarily due to their size. It is well known that reactivity increases as particle size decreases. Size was considered as part of the covalent functionalization strategy, as reducing it increases the available surface area for functionalization. Also, they were the ones that we have in the laboratory for a parallel experiment.

  1. The long-term stability and reusability of the nanosensor are not discussed.

R = We thank Reviewer 2 for this comment. Our aim in developing this work was to present a preliminary assay on the detection capability of the synthesized hybrid nanosensor without intending to portray the work as the development of a fully validated sensor. Therefore, we adjusted our focus to highlight our main objective: the covalent functionalization of SiO2 nanoparticles with a molecule specifically designed to bind them through covalent bonds. Consequently, this preliminary study did not include a discussion on the long-term stability and reusability of the nanosensor, as we aimed to demonstrate the viability of covalent functionalization for its subsequent application in analyte detection.

  1. Why kerosine is selected?

R = We thank Reviewer 2 for this comment. Kerosene was chosen because its presence has been documented in numerous accidents and spills both in our country (Chile) and in other parts of the world. For example, in 2017, an incident occurred at the La Parva Ski Resort, where approximately 15 thousand liters of kerosene were spilled into the Mapocho River, a crucial water body in the Metropolitan Region of Chile, directly affecting the biodiversity of the environment.

news; https://www.df.cl/mercados/pensiones/aseguradoras-se-enfrentan-a-la-parva-por-derrame-de-parafina-en-rio-mapocho

  1. The concentration used for the detection of kerosine was not optimized.

R = We thank Reviewer 2 for their comment. We apologize if our main focus was not clear; however, we have made modifications to the manuscript to emphasize that the novelty lies in the covalent modification of nSiO2 nanoparticles with curcuminoid molecules to generate a new hybrid nanosensor. We presented a preliminary assay solely to corroborate its capability as a detection probe. As part of future work, parameters will be evaluated to validate its application as an organic liquid sensor, including the optimization of kerosene concentration.

Round 2

Reviewer 1 Report

Comments and Suggestions for Authors

I am satisfied with the revised version

Reviewer 2 Report

Comments and Suggestions for Authors

Nil